# Optical Coherence Tomography Angiography in Diabetic Patients: A Systematic Review

**DOI:** 10.3390/biomedicines10010088

**Published:** 2021-12-31

**Authors:** Ana Boned-Murillo, Henar Albertos-Arranz, María Dolores Diaz-Barreda, Elvira Orduna-Hospital, Ana Sánchez-Cano, Antonio Ferreras, Nicolás Cuenca, Isabel Pinilla

**Affiliations:** 1Department of Ophthalmology, Lozano Blesa University Hospital, 50009 Zaragoza, Spain; abonedm@salud.aragon.es (A.B.-M.); mddiaz@salud.aragon.es (M.D.D.-B.); 2Department of Physiology, Genetics and Microbiology, University of Alicante, 03690 Alicante, Spain; henar.albertos@ua.es (H.A.-A.); cuenca@ua.es (N.C.); 3Aragón Health Research Institute (IIS Aragón), 50009 Zaragoza, Spain; eordunahospital@unizar.es (E.O.-H.); anaisa@unizar.es (A.S.-C.); aferreras@unizar.es (A.F.); 4Department of Applied Physics, University of Zaragoza, 50009 Zaragoza, Spain; 5Department of Ophthalmology, Miguel Servet University Hospital, 50009 Zaragoza, Spain; 6Department of Surgery, University of Zaragoza, 50009 Zaragoza, Spain

**Keywords:** diabetes mellitus, diabetic retinopathy, foveal avascular zone, FAZ, optical coherence tomography angiography, OCTA, diabetic macular oedema

## Abstract

Background: Diabetic retinopathy (DR) is the leading cause of legal blindness in the working population in developed countries. Optical coherence tomography (OCT) angiography (OCTA) has risen as an essential tool in the diagnosis and control of diabetic patients, with and without DR, allowing visualisation of the retinal and choroidal microvasculature, their qualitative and quantitative changes, the progression of vascular disease, quantification of ischaemic areas, and the detection of preclinical changes. The aim of this article is to analyse the current applications of OCTA and provide an updated overview of them in the evaluation of DR. Methods: A systematic literature search was performed in PubMed and Embase, including the keywords “OCTA” OR “OCT angiography” OR “optical coherence tomography angiography” AND “diabetes” OR “diabetes mellitus” OR “diabetic retinopathy” OR “diabetic maculopathy” OR “diabetic macular oedema” OR “diabetic macular ischaemia”. Of the 1456 studies initially identified, 107 studies were screened after duplication, and those articles that did not meet the selection criteria were removed. Finally, after looking for missing data, we included 135 studies in this review. Results: We present the common and distinctive findings in the analysed papers after the literature search including the diagnostic use of OCTA in diabetes mellitus (DM) patients. We describe previous findings in retinal vascularization, including microaneurysms, foveal avascular zone (FAZ) changes in both size and morphology, changes in vascular perfusion, the appearance of retinal microvascular abnormalities or new vessels, and diabetic macular oedema (DME) and the use of deep learning technology applied to this disease. Conclusion: OCTA findings enable the diagnosis and follow-up of DM patients, including those with no detectable lesions with other devices. The evaluation of retinal and choroidal plexuses using OCTA is a fundamental tool for the diagnosis and prognosis of DR.

## 1. Introduction

Diabetes mellitus (DM) is a metabolic disease caused by an increase in glucose levels due to a diminution of insulin secretion or an increase in resistance to its activity. DM is expected to increase worldwide in a rapid manner, increasing by 25% by 2030 and 51% by 2045 [1].

Diabetic retinopathy (DR) is the most severe and frequent ophthalmic complication [1,2]. DR is the leading cause of legal blindness in the working population in developing countries. Diabetic patients may primarily have a neurodegeneration process in the retina, followed by well-known changes at the microvascular net, with pericyte loss and an increase in thickness of the basal membrane, with a breakdown of the inner blood retinal barrier. Although there is a trend towards a reduction in proliferative DR incidence due to improved control, the prevalence of DR in DM patients is approximately 35% of that in other patients [3,4,5,6]. There was a period with no DR signs but neuronal degeneration and microvascular changes, which were not detectable by ordinary examination. Any method that could help to find changes before these changes are evident would be a fundamental tool as a disease biomarker [7].

The Early Treatment Diabetic Retinopathy Study (ETDRS) divided DR into groups: nonproliferative DR (NPDR) and proliferative DR (PDR). NPDR can be divided into mild, moderate, and severe DR [8]. At each level, patients present different changes in their fundus related to either an increase in vessel permeability or a diminution of the vascular supply, including microaneurysms (MAs), exudation or oedema, haemorrhages, intraretinal microvascular abnormalities (IRMAs), vascular changes, and neovascularization (NV) [9].

Clinical examination of diabetic patients was based on ophthalmoscopy under mydriasis, fundus photography, fluorescein angiography (FA), and optical coherence tomography (OCT).

FA has been considered the gold standard to evaluate retinal vascularization [10]. FA can evaluate vascular integrity, the presence of MA, the loss of vascular perfusion, and the increased permeability of the vessels generating oedema and NV. FA is an invasive test that can generate severe adverse effects in a small percentage of patients [11]. Due to its potential secondary effects, it is not usually performed as a screening method.

The development of OCT angiography (OCTA) has been an important tool in the control of DM patients with and without microvascular lesions.

### 1.1. Morphology of the Retinal and Choroidal Blood Vessels

The central retinal artery, a terminal branch of the ophthalmic artery, divides and forms the different retina plexuses covering the entire retina, excluding the central foveal avascular zone (FAZ) and the most peripheral 1–1.5 mm [12,13,14,15]. The number of retinal plexuses varies from one to four depending on the eccentricity [16]. These plexuses are the radial peripapillary capillary network (RPCN), close to the optic nerve head, the superficial capillary plexus (SCP), the intermediate capillary plexus (ICP), and the deep capillary plexus (DCP) [17]. The rest of the central retina is formed by the SCP, ICP, and DCP [17] (Figure 1). For more information about retinal morphology and vascularization, see Appendix A.

### 1.2. Optical Coherence Tomography Angiography (OCTA)

OCTA is a new non-invasive angiography technique based on OCT technology that can be performed without pupillary dilation. It provides high-resolution images of the retinal capillary plexuses and the choriocapillaris (CC) without using any contrast in a rapid and easy way. Using more advanced hardware and acquisition software, OCTA enables greyscale retinal vascular flow imaging. It can provide perfusion density maps and average perfusion density. OCTA is able to visualise changes in the DR, such as MA, nonperfusion areas, IRMA, or NV. OCTA can demonstrate noncapillary perfusion areas even better than FA, and the image will not have interference from any leakage [18,19]. It is based on the detection of moving blood cells, such as red blood cells. Performing consecutive B-scans in the same location on the retina shows the presence of movement through the blood vessels. The change in contrast over time indicates the vessel location and erythrocyte movement through them [20]. Changes are subsequently processed with different computer algorithms. This technology allows en face images and reconstructions of different retinal layers to be obtained. Image capture in OCTA requires great precision because it is based on differential analysis of the B-scan changes related to erythrocyte micromovements and the high sequential image speeds of the devices [20].

The maps generated by OCTA are a representation of retinal vascularization over a particular area of interest, in this case the macular area, and according to different anatomically interesting segmentation profiles. In OCTA, the introduction of projection resolved OCTA algorithms and three-dimensional visualisation increased the depth quality of the images. OCTA allows retinal segmentation into different vascular and nonvascular layers: SCP, ICP, DCP, external retina, and CC. The introduction of wild field (WF) OCTA allows better knowledge of the capillary plexuses in both the mid- and far periphery. The in-depth resolution enables visualisation of aneurysmal dilatations in the plexuses, avascular or low perfusion areas, retinal NV or intraretinal shunts, vascular structures in the choroid, or loss of CC vessels.

OCTA provides a dye-free image useful to detect angiographic signs of DR and changes in the capillary network at the macular level, even before onset of the disease. In patients with DR, areas of nonperfusion and their location in the SCP and DCP, as well as irregular capillaries, MA dilatations, and modifications in the CC layer, have been clearly analysed [21,22]. In addition to these qualitative characteristics, OCTA can provide a quantitative analysis of the density and flow of retinal blood vessels in each layer [23].

OCTA disadvantages include the loss of findings in which flow is slow, the inability to see leakage or staining, and difficulty visualising the peripheral retina. OCTA is an effective tool to evaluate DR, but the large amount of data and protocols can generate problems in the most sensitive parameters [24].

In summary, OCTA data acquisition is faster than FA and is three-dimensional and depth-resolved, allowing individual capillary plexuses automatically assessment based on current software algorithm. OCTA allows the visualisation of all plexuses, including the intermediate capillary, detecting pathological features that are not available in traditional dye-based angiography. In addition, as a non-invasive and rapid test, it is adequate for patients who require frequent follow-up exams. Nevertheless, FA is still the gold standard for retinal vessel evaluation, providing some additional findings such as leakage.

The purpose of this review was to provide an actual summary of the different findings assessed by OCTA and the diagnostic value of OCTA in DR patients, which is a great future challenge due to the prevalence of DM and the heavy burden caused by DR.

## 2. Methods

### 2.1. Literature Search

A systematic review was performed following the Preferred Reporting Items for Systematic Reviews and Meta-Analyses (PRISMA) guidelines [25] using a PRISMA checklist. It included a comprehensive search of different databases, including PubMed and EMBASE, last run-on 15 April 2021, for the following terms: OCTA OR OCT angiography OR optical coherence tomography angiography AND diabetes OR diabetes mellitus OR diabetic retinopathy OR diabetic maculopathy OR diabetic macular oedema OR diabetic macular ischaemia including MeSH terms and synonyms.

### 2.2. Inclusion/Exclusion Criteria

The search was performed to identify those studies in which OCTA was used to image diabetes patients with or without any type of DR. The included studies were limited to those published in English and in peer-reviewed journals, excluding case reports, conference proceedings and letters, and studies based on time-domain OCT. No restrictions existed for age, diabetes type or control, or follow-up.

### 2.3. Literature Review

Using the search criteria described above, a total of 829 results in PubMed and 627 in Embase, a total of 1456 records were found. PRISMA search was performed by three authors. After an initial review of abstracts by two independent reviewers, removal of duplicate studies or those articles that did not meet the selection criteria, 107 articles were selected for a full literature review. Other papers previously cited that were not selected, which appeared to be important to our review, were added supported by a third author, and at the end, a total of 135 studies were included in this qualitative systematic review. OCTA has been used to evaluate any kind of change in DM patients with or without DR. We described selected paper findings in the FAZ, MA, nonperfusion areas, ischaemia, IRMA, NV, and diabetic macular oedema (DME) (Figure 2).

Each of these topics will be discussed in turn, followed by a discussion of OCTA’s future directions in DR.

## 3. Results and Discussion

### 3.1. Foveal Avascular Zone

Different papers have analysed FAZ changes in either diabetes mellitus type 1 (DM1) or diabetes mellitus type 2 (DM2) with or without DR. FAZ has previously been analysed using FA, only evaluating changes at the SCP. FA showed that FAZ increased in DR with retinopathy stage [26] due to loss of the surrounding capillary [27]. Studies have demonstrated that compared with FA, OCTA allows better discrimination of the central and parafoveal macular microvasculature, especially for FAZ disruption and capillary dropout [28] (Figure 3A,B).

Nevertheless, not all OCTA studies in DM had the same results evaluating the FAZ, and the methodology differed between them, evaluating the plexuses one by one or all in total. Other discrepancies between studies were in the way they dealt with projection artefacts or artefacts caused by vitreous opacities [29].

The first author who analysed OCTA changes in patients with no DR signs was De Carlo [30], who demonstrated that DM without DR showed an increase in the FAZ and areas of capillary nonperfusion (considering both DM1 and DM2 patients). Similarly, Dimitrova and colleagues showed an increase in the FAZ of the SCP in DM patients without DR as well as a decrease in vessel density (VD) in both plexuses [31] (Figure 4). 

Takase et al., found an increase in the en face FAZ in all DM patients with or without DR signs [32], and Di et al., described an increase in FAZ in DM patients. They found that patients with more severe retinal damage had a much larger FAZ, with changes in the area and vertical and horizontal radius [33]. An enlarged FAZ associated with a reduction in the VD of the SCP and DCP in the foveal and parafoveal areas has been observed in patients with NPDR [34]. Comparing DM1 patients without DR or with mild NPDR with controls, Simonett et al., suggested that parafoveal capillary nonperfusion in DM1 is an early marker of retinal changes starting in the DCP [35]. Wang et al., as previously described, postulated that FAZ metrics may have a prognostic value in DR progression, DME, and visual acuity (VA), but highlighted that the high variation among normal individuals in FAZ area and perimeter makes them less than ideal biomarkers for staging DR. None of their FAZ metrics differed with the severity of DR, indicating that they may not play an important role in advanced DR, but may have a prognostic role in predicting DR progression, DME, and VA [36].

Parafoveal nonperfusion has been analysed using different strategies to identify a better biomarker for DR severity [23,37,38,39] and ischaemic index [40]. When the DR appeared, this density changed into a progressive loss of the capillary network. Xu and You [29,41] indicated that FAZ and nonperfusion areas were both significantly larger in the diabetic group, whereas the FAZ circularity was significantly smaller [41]. You et al. also demonstrated that treatment requirements were related to the extrafoveal avascular areas for the baseline DCP [29]. The nonperfusion ratio was studied by Wang et al., who found a significantly lower parafoveal VD in DR patients compared with those without DR, with an increase in VD loss related to DR severity [42]. VD diminished with age and higher HbA1c levels, and patients with DME had a significantly lower average parafoveal VD according to Xie and colleagues [43]. Other authors, such as Rosen, found an increased area of capillary density in DM patients without DR after extracting the noncapillary structures, suggesting an autoregulatory response to an increase in metabolic needs [44], highlighting that OCTA may help identify early-stage DR before retinopathy is apparent. Rosen suggested that perfused capillary density is a more sensitive marker to detect differences between healthy individuals and DM patients than FAZ metrics [44].

In contrast, some studies deny OCTA as the most appropriate tool for detecting preclinical changes in patients with diabetes, suggesting that clinical examinations and glycaemic control should be kept on as the primary clinical parameter during DR screening [45].

Differences in FAZ parameters between DM1 and DM2 patients have been studied. Oliverio et al. found that changes in FAZ parameters were more pronounced in DM1, and these modifications were correlated with the duration of the disease [46]. Vujosevic and Um indicated that the increase in FAZ area, and a decrease in VD, are related to DR progression and are more severe in the DCP than in the SCP in both DM types [47,48].

Changes in the FAZ can be related to visual impairment [49,50], as was demonstrated in Samara’s study. They found a negative correlation between logMAR VA and VD in both SCP and DCP, and a positive correlation between logMAR VA and FAZ area in both plexuses [51].

FAZ morphology can be visualised in en face projections. The SCP is formed by large and small capillaries that end at the FAZ as a terminate capillary ring with a centripetally branching pattern. The DCP ends at the macula with lobular patterns with no direction [16]. The acircularity index provides information about the extent to which the FAZ differs from a circle. Krawitz and colleagues found differences in the FAZ shape between controls and all DR patients, but not DM patients with no lesions. The mean acircularity index was 1.32 in both the control and no DR groups, 1.57 in the NPDR group, and 1.78 in the PDR group. There were no differences between NPDR and PDR [52]. They also considered the axis ratio as an index of disease progression and therapeutic interventions. The average axis ratios were 1.17, 1.12, 1.27, and 1.33 in the different stages. A higher acircularity index and axis ratio were associated with a worse stage of DR.

Zahid and colleagues [53], using fractal dimension (FD) analysis, a mathematical method to evaluate the complexity of tissues, found a reduction in the flow in DR, both in SCP and DCP, in the absence of DME. Tang et al. also observed a lower FD associated with DR severity and an increased FAZ area and decreased FAZ circularity [54]. The study performed by Sun et al. evaluated the risk of DR progression and DME development beyond traditional risk factors and related FAZ area, VD, and FD of DCP with DR progression, whereas VD of SCP would predict DME development [55]. Other authors suggested FD-300 analysis (VD of a 300 μm width annulus surrounding the FAZ) was useful for detecting preclinical microvascular alterations in DR screening [56].

In addition, the decrease in FAZ circularity and parafoveal vessel density are postulated to be related to structural retinal neurodegeneration, because they would be highly correlated with ganglion cell layer—inner plexiform layer (GCL-IPL) thinning, regardless of the presence of DR, and would predict microvascular impairment in early DR [57,58].

#### 3.1.1. Microaneurysms

Microaneurysms (MAs) were identified using OCTA (Figure 3A,B). We found saccular dilatation or fusiform capillaries, as described by Ishibazawa et al. [18], in both the SCP and DCP. According to Park and colleagues [59], it is also possible to identify MAs in the ICP. OCTA is able to identify a smaller number of MAs than FA, but it has the ability to detect MAs in both the SCP and DCP (Figure 5).

Salz and colleagues found that compared with FA, OCTA had a sensitivity of 85% (95% CI, 53–97%) and a specificity of 75% (95% CI, 21–98%) in detecting MAs. [60]. These results have been supported by other studies [19,61]. Soares and colleagues compared FA and OCTA AngioVue and AngioPlex, with FA being superior to both for detecting MAs in both the SCP and total retina slab [28]. As already described, MAs were more frequently located in the DCP. MAs were related to ischaemic areas, and they found MAs surrounding nonperfusion areas (NPAs) [19]. Parrulli et al. also found that FA is the best way to detect MAs. OCTA devices can differentiate their detection depending on the number of B-scans, with great variability between devices [62]. Hamada et al. also found discrepancies between FA, OCT B-scan, and OCTA, with the latter being able to overlook MAs in patients with diabetic macular oedema (DME) [63].

Park et al. described MAs in all three plexuses [59]. Other authors found a higher number of MAs in the DCP than in the SCP [18,19,64]. Byeon et al. described the deep location of the MA [65] leaking in the outer plexiform layer (OPL). MAs can protrude towards outer layers, such as the outer nuclear layer (ONL).

Some authors have correlated the MAs on structural OCT and OCTA. Parravano studied the correlation between them and their evolution [64,66]. They described two MA patterns based on OCT findings. Hyporeflective lesions on structural OCT were less visualised using OCTA than hyper-reflective or moderate lesions (66.7% vs. 88.9%). They suggested that the hyporeactive lesions could have a lower flow that was not detected with OCTA. Other authors have suggested the possibility of turbulent flow [18], or that MAs are not perfused with luminal fibrosis and lipid infiltration in their histology [67]. Parravano [64,66] also described the different behaviours depending on their OCT reflectivity. MAs that developed over 12 months in extracellular fluid were hyper-reflective (66% vs. 18% of the hyporeactive MAs). The location was also related to fluid development: those located in the DCP were those which leaked after one year. The relationship of DCP MAs with the development of DME has also been described by Hasegawa et al. [68]. In summary, 12 months after their description, MAs with a hyper-reflective pattern persisted on OCTA and were mainly located in the DCP.

Schaal et al., studied the agreement in the detection of DR signs on colour fundus photography (CFP) versus SS-OCTA. In patients with an ETDRS level ≥ CFP, MAs were found in 90% of the cases, close to 91% of which were found with OCTA. They suggested that MAs are more apparent on the 3 × 3 mm scan than on the 12 × 12 scan due to the lower resolution [69]. Following their assessment of WF-OCTA, Tian and colleagues used 12 × 12 mm scanning with different slabs for MAs. They analysed 247 eyes of patients with DM and detected MAs in 60.6% of the eyes using the retinal slab and 59.8% in the SCP slab, with no significant differences between them. No MAs were evaluated in the deep slab because of the poor details. This study provides similar results in relation to IRMAs [70].

Carnevalli and colleagues did not find MAs in their DM1 patients or other anatomical changes. Their population was young (mean age 22 ± 2 years) and had a short disease duration (11 ± 4 years) [71]. They only found rarefaction of the perifoveal capillary network in the SCP in 28% of their series. Park and colleagues evaluated the microvascular changes in the foveal and parafoveal areas in 64 patients with NDR and a mean age of 61.0 ± 9.34 years. They identified MAs in only 9.38% of the cases. They found no association between changes in VD occurring in the different plexuses, disease duration, best corrected visual acuity (BCVA), FAZs, or analytical parameters (HbA1c, serum creatinine or e-GRF) [72].

In 2019, Thompson et al., found MAs in 60% of their patients, a small sample of DM2 with good glycaemic control and without DR signs [73].

#### 3.1.2. Nonperfusion Areas

Loss of vascular perfusion is an indication of ischaemia. OCTA can evaluate both macular and peripheral retinas using WF strategies, including WF-OCTA. Macular ischaemia is related to VA in DM patients. NPAs were evaluated using automated quantification of the VD or the total area of vessel nonperfusion and with the FAZ diameter and changes.

FA shows NPA between the large retinal vessel. OCTA detected NPAs not only in the SCP, but also in the ICP and DCP (Figure 3C–F). OCTA clearly visualised the border between sparse capillary areas and dense capillary areas, with a sensitivity of 98% and specificity of 82%. Therefore, OCTA is a better procedure to detect capillary density than conventional FA [18,19,74]. This capillary density reduction is associated with remodelling and enlargement of the FAZ even before MAs, which are currently believed to be the first clinical sign of DR [75]. De Carlo et al. [30] reported changes in the FAZ (increased FAZ area and the presence of FAZ remodelling) and capillary nonperfusion in patients with DM with no signs of DR.

Loss of perfusion has been described in all plexuses, including the ICP [76]. Onishi et al. suggested a significant increase in NPAs in all three plexuses in the NPDR group compared with controls. Zhang et al., also identified a significant increase in superficial NPAs in DM patients without DR compared with controls [74,76]. The authors emphasise the importance of OCTA segmentation schemes that consider the ICP separately from the SCP and DCP. Simonett et al. [35], studying patients with DM1 and without DR or with mild NPDR, reported a decreased parafoveal VD (similar to parafoveal capillary nonperfusion) only in the DCP, with no changes in the FAZ area in either the SCP or DCP. Dimitrova et al. [31] documented a decreased parafoveal VD in the SCP and DCP and an increased FAZ area in the SCP in patients with DM (mostly in DM2) and no DR compared with control subjects. Choi et al. [77] documented retinal microvascular abnormalities (including capillary dropout, dilated capillary loops, tortuous capillary branches, patches of reduced capillary perfusion, irregular FAZ contours, and/or FAZ enlargement) in all 3 plexuses in 18 of the 51 eyes with DM and no clinical signs of DR (with no specification of DM type). Moreover, these authors reported focal or diffuse CC flow impairment in almost half of the evaluated patients without DR.

DM patients showed a progressive diminution of the capillary density with the severity of the ocular manifestations [28]. Severe NPDRs showed an increase in NPAs [78], enlarged spaces between the large and small vessels in the SCP and DCP with an increased and irregular FAZ. IRMAs and NVCs were more frequently associated with NPAs [79]. Vujosevic et al. indicated that both SCP and DCP are prematurely altered in patients with DM1 and without clinical signs of DR, whereas in patients with DM2, the DCP is the first affected plexus [48].

Although there are different results, NPAs and capillary dilation have been described as more prominent in the SCP, and MA is common in the DCP. Studies have revealed that the severity of vascular changes in the SCP is closely related to abnormalities in the SCP [78].

Nesper et al., described the percentage area of the retina and CC related to NPA. NPA was significantly correlated with disease stage when considering retinal vascular changes, but no significant correlation was found for CC [75,80].

Yasukura et al. studied differences between macular and extramacular NPAs related to arterial distribution [81]. They did not find differences in extramacular NPAs between severe NPDR and PDR. Eyes with PDR had significantly greater NPAs in the macular area than those with severe NPDR. They suggested that the extramacular region between two arteriolar branches is the most vulnerable to DM capillary loss [81].

OCTA demonstrated impaired perfusion within cotton-wool spots [21]. Extramacular cotton-wool spots (or white spots) are mostly associated with NPAs encompassing all retinal layers, in contrast to macular cotton-wool spots that are more associated with NPAs in the superficial layer only [80].

WF-OCTA demonstrates preferential location of NPAs along the main retinal arteries in all stages of DR. Tan et al., using 12 × 12 WF-OCTA, found an increase in capillary dropout in the peripheral annulus that increases with the severity of DR. They suggested that the capillary dropout density in the peripheral subfield is the best parameter to discriminate between mild NPDR and DM patients without DR [82]. Diabetic microangiopathy is a midperipheral disease and firstly affects the temporal quadrants. The midperiphery has a smaller vascular supply (vascular plexuses merged from three to two plexuses), and the nasal quadrants are supplied by the radial peripapillary capillary plexus [83]. Ishibazawa et al. studied NPAs with OCT images and found that NPAs were more frequently adjacent to arterial vessels. They hypothesised that diabetic microangiopathy started near the arterial side, with no regard to the level of DR severity, and then progressed towards the venous side [84].

In addition, more pronounced vascular involvement in the DCP has been described, regardless of the stage of DR, which may be explained by the difference in the perfusion pressure between the SCP and the DCP [85].

Thus, the advantage of OCTA lies in its ability to detect both peripheral retinal nonperfusion and eventual peripheral active NV, which remains difficult to visualise clinically.

#### 3.1.3. Ischaemia

DM patients exhibit a reduction in capillary density. Diabetic macular ischaemia (DMI) is associated with an enlargement and disruption of the FAZ and with retinal capillary dropout in noncontiguous areas of the macula, providing important clinical and prognostic information regarding disease severity and predicting DR progression [52,86,87,88]. Similar to FA, OCTA is capable of grading and quantifying DMI through several OCTA parameters, such as the perifoveal intercapillary area, total avascular area, or extrafoveal avascular area [89]. OCTA is better at detecting capillary density than conventional FA [18,19]. One of the key advantages of OCTA over FA is the ability to noninvasively detect DMI [90]. OCTA could even identify DMI in eyes with relatively few symptoms. The FAZ area in both the SCP and DCP increased with DR severity, and the FAZ area (at SCP) correlated with retinal sensitivity at baseline [91]. However, it remains unclear whether eyes with DMI detected by OCTA have higher risk of progressive visual loss the OCTA findings continue to deteriorate over time. Loss of perfusion has been described in all plexuses. Authors agree that grading DMI in the three plexuses (SCP, ICP, and DCP) had a higher sensitivity and specificity for determining DR stage and comparing DR versus healthy controls [59] than full retinal angiograms [92]. Changes in the ICP are important to consider [76].

Significant deterioration in OCTA parameters over time in DR patients has been described. Kim et al. [57] recently reported that microvascular impairment is progressive even in early stages of DR. They observed SCP VD loss in 40 eyes with no DR or mild NPDR over a period of two years. However, DCP VD was not studied in this report [57]. Tsai et al. [91] observed a significant deterioration in the DCP parafoveal VD and SCP FAZ area in patients with various severities of DR at baseline over a one-year follow-up period. A larger DCP FAZ area at baseline was associated with a significant worsening of BCVA over one year. Similar to previous studies, Xie et al. [43] and Ragkousis et al. [56] observed a decrease in parafoveal vascular density as the disease progressed. In this sense, in addition to the density of the DCP, other parameters, such as the VD of the extrafoveal avascular area and the vessel length fraction of the DCP, appear to decrease with the severity of the disease [36]. Specifically, the vessel diameter index at the SCP and the VD in the DCP showed the best correlations with the severity of DR [36]. This finding not only provided evidence that OCTA parameters are able to predict visual outcomes in DR, but also suggests the importance of the detection and monitoring of DCP parameters in ischaemic conditions such as DR, as previously described.

Previous cross-sectional studies have correlated central visual loss in diabetic eyes with the degree of parafoveal capillary loss [50,51]. Such relationships are more prominent with alterations in the DCP than in the SCP [93,94]. Changes in the DCP have also been found to correlate better with DR severity than changes in the SCP [76]. These observations are supported by histologic studies that show higher vulnerability of the deep foveal plexus to endothelial injury [95]. Furthermore, there have been many reports highlighting DCP ischaemia, which is an important finding in DR. Scarinci et al. [96] suggested that nonperfusion of the DCP is associated with photoreceptor disruption in DMI. The flow density of CC in patients with severe DR seems to be associated with the severity of the disease because the flow decreases as the disease worsens [97]. Lee et al. [98] showed that a poor response to anti-VEGF agents in DME is associated with DCP damage but not SCP damage. An increase in DCP destruction with DR progression was also reported [99]. Early DCP vascular alterations were found, especially in DM1, which were evident even before the diagnosis of clinically detectable DR [71,100]. Moreover, VD in the fovea, parafovea, and peripapillary area and the flow area in the choroid was also reduced in DM2 patients without signs of DR [101,102]. The FAZ area and VD change more rapidly as DR progresses in the DCP than in the SCP during the progression of DR [47].

Tsai et al. [91] also demonstrated the predictive value of structural OCTA parameters in relation to visual outcomes beyond current established systemic risk factors. Larger baseline FAZ areas in the DPC were associated with worsening visual outcomes, and larger decreases in SCP VD were associated with worsening retinal sensitivity over one year. These associations support the use of OCTA in the early detection and monitoring of DMI. Additionally, it has been demonstrated that OCTA parameters such as larger FAZ areas and lower VD in DCP at baseline increase the likelihood of DR progression within two years [55].

Cao et al. found that OCTA can be a useful way to identify preclinical lesions in DM2 based on capillary perfusion. They found a decreased vessel density in the SCP, DCP, and CC in DM2 patients before having any DR signs, without changes in the FAZ area [103]. When analysing capillary perfusion in SCP, DCP, and CC in the 3 × 3 mm and 6 × 6 mm protocols, diabetic patients had significantly lower perfusion than the control group. Normal subjects had higher capillary perfusion rates than patients diagnosed with mild nonproliferative DR (NPDR) [104,105].

The identification of DCP by OCTA plays an important role in DMI. Minnella et al. studied eyes with DMI and demonstrated significantly increased perifoveal “no flow” areas in both the SCP and DCP compared with controls, and Scarinci et al. found that these areas of DCP nonperfusion corresponded precisely with areas of outer retinal disruption on structural OCT imaging [96,106].

Furthermore, vascular complexity and morphology are evaluated by combining VD, fractal dimension, and vessel diameter and allow to define the state of DR. VD, defined as the ratio of blood vessel area to the total measured area, decreases in both SCP and DCP in patients with DR and in diabetic patients without DR [78,107].

Taewoong et al. [47] showed a deterioration in the FAZ area and VD in the SCP and DCP as DR progressed in both DM1 and DM2, similar to previous reports [23,108]. This deterioration was more prominent in DCP than in SCP, regardless of the diabetic type. However, in DM1, the deterioration of VD was delayed until the DR reached a severe NPDR stage, whereas there was a gradual decline in VD in DM2. This finding may be caused by differences in the pathophysiology of the two types of diabetes and may explain the different clinical manifestations [47].

Compared with eyes with mild and moderate NPDR, eyes with severe NPDR and PDR demonstrated a significant decrease in VD [86,109]. The superficial capillary network supplies the ganglion cell complex and inner nuclear layer; thus, a decreased superficial capillary network and loss of the ganglion cell complex have been detected [110].

Authors such as Wang et al. [111] and Agemy et al. [86] showed statistically significant reductions in VD in diabetic patients compared with controls using different approaches to calculate vascular density as a trend towards reducing vascular density with worsening severity of diabetic disease. Thus, the analysis of vessel density is associated with the degree of disease and risk of DR progression and may be a useful potential predictor of proliferative DR.

On the other hand, FD represents vascular complexity and microvascular morphology related to macular ischaemia in both SCP and DCP. Zahid et al. studied this entity and observed, DR patients showed a decrease in vascular density and increased fractal dimension with a greater average vascular calibre secondary to hypoxic conditions [7,53]. Ting et al. studied the capillary density index and FD in DM2 patients and reported a decrease in both SCP and DCP capillary density with DR progression and an increase in FD in both plexuses [108]. Moreover, receiver operating characteristic (ROC) curve analysis defined skeletonised FD, vessel length density, and vessel diameter index as the most effective parameters to detect glycaemic changes in DM2 patients [105].

FA images use the ETDRS protocols to grade DMI as follows: absent (no FAZ disruption), questionable (FAZ not smooth/oval, but no clear pathology), mild (<half FAZ circumference destroyed), moderate (>half FAZ circumference destroyed), severe (FAZ outline destroyed), or ungradable [10].

Several studies have compared OCT to the FA grading of DMI. Bradley et al., studied the reproducibility of the OCTA-based grading. SCP OCTA images were graded using the ETDRS protocols [88] and compared with FA images, at the DCP, this grading was absent (no disruption of FAZ), questionable (FAZ not smooth/oval, but no clear pathology), mild/moderate (FAZ disrupted in ≤2 quadrants), severe (FAZ disrupted in ≥3 quadrants), or ungradable (poor image quality, artefact). CC was graded as ischaemia present (loss of speckled hyper-reflectance or dark defects), ischaemia absent, or ungradable, and obtained substantial intergrader agreement in terms of the DMI grade acquired for the SCP, DCP, and CC [86].

#### 3.1.4. Intraretinal Microvascular Abnormalities

OCTA can detect IRMA as intraretinal looping vessels of capillary origin with a larger calibre than the surrounding vessels with a flow that does not cross the internal limiting membrane (ILM), and they are usually located in areas with little or no perfusion [21,112] (Figure 3D,E). In contrast, NVE passes the ILM and protrudes into the vitreous cavity [85]. Matsunaga et al. described an increase in the calibre of the loops compared them to surrounding capillaries [21], and they described one case whose origin was a vessel with a large diameter [21]. Their characteristics make it difficult to detect them in CFP [69]. Schaal and colleagues found that OCTA had a higher detection rate than CFP. They analysed two cohorts of diabetic eyes in a retrospective cross-sectional observational study, one using SS-OCTA grading and the other comparing OCTA and CFP. In patients with ETDRS severity levels over 43, OCTA was able to detect a significantly higher number of IRMAs (85% of the sample) than CFP (detected in only 35%). The inter-device agreement was only fair, with k = 0.2. They suggested that the presence of adjacent areas of capillary dropout helps to identify IRMA detection with OCTA compared with CFP. These researchers also compared three different slabs with 12 × 12 WF-OCTA to detect DR findings. IRMAs were more frequently detected on the retinal slab but with no differences from the SCP slab [70]. Compared with FA, according to Arya et al. [113], OCTA achieved 99% specificity and 92% sensitivity. It might even be more accurate pictures, as it has no dye leakage that appears with FA and its ability to segment layers [112].

Furthermore, Cui et al. published a study [114] showing that ultra-widefield OCTA (UW-OCTA) was superior in the number of IRMAs detected per ultra-widefield CFP (UW-CFP) (*p* < 0.001) and had almost 100% agreement (k = 0.916) with ultra-widefield FA (UW-FA).

Regarding their distribution, Tian et al. [70] found no significant differences between the retinal and SCP slabs, detecting none in the SCP slab using the 12 × 12 mm swept source OCTA (SS-OCTA) protocol.

The utility of OCTA in the follow-up of DR patients with IRMA before and after different therapies has also been evaluated. Sorour et al. [115] used it to study the structural changes in 45 IRMAs after anti-VEGF treatment compared with patients with similar DR who did not receive treatment. At the baseline visit, they characterised different morphologies (dilated trunk, loop, pigtail, sea-fan-shaped, and net-shaped) with higher complexity and more advanced pathology. However, they found no relationship between them, the number of injections and their response to treatment. The quantity of IRMA detected with OCTA has also been related to the severity of DM according to Kaoual et al. [116].

Shimouchi et al. [117] conducted a retrospective study in 46 eyes of 29 patients proposing a classification to standardise the changes after panretinal photoagulation (PRP). They established five groups: unchanged, tuft regression, repercussion, mixed, and worsening [118]. Those IRMAs that regressed were adjacent to areas of restored perfusion after PRP [117]. Russell et al. [119] also focused on changes after PRP in a prospective study of 20 patients. They found how four IRMAs detected in two different patients by FA, OCTA, and B-scan progressed to NV. The description of IRMAs as precursors of NV, although controversial, has been proposed and described in other investigations [120,121,122].

#### 3.1.5. Neovascularization

Retinal NV is one of the key signs of PDR responsible for vision loss. Thus, early detection could improve visual prognosis [123]. Fundus eye exam, OCT, and FA have been used to identify NV [121]. FA has always been the gold standard to analyse NV. However, early leakage in FA prevents the exact assessment of NV areas, which can already be seen with OCTA [19,30,124]. OCTA imaging has become a useful tool for NV diagnosis [121]. OCTA can identify NV arising at the optic nerve (NVD) or in other retinal places (NVE). OCTA was also able to estimate NV activity (Figure 3F).

OCTA detects changes prior to the appearance of neovascularization and the presence of NV and assesses its progression, either in active or fibrotic NV. Onishi et al. discussed how the vascular changes “precursors of neovascularization” observed in the superficial plexus (dilatation, telangiectasia with high flows) can lead to a “steal phenomenon” increasing ischaemic phenomena in deeper plexuses [76].

Different authors have classified active NV by evaluating either the morphology of the NV and its origin [121,125] or blood flow and density maps [126]. One of the first descriptions of NV was given by Ishibazawa et al., characterising two different patterns both at the optic nerve and in other retinal places: one included exuberant vascular proliferation with small and irregular new vessels, and the other was described as pruned NVs that did not show leakage at the FA [125]. Elbendary et al. categorised active NVD according to their morphology on OCT, OCTA, and B-scan [126]. Blood flow data and density maps were the main features used to determine NV activity. They defined three different patterns in disk NV depending on the blood flow observed in OCT (vascular, fibrovascular, or fibrous component), and two in NV elsewhere, branching vascular tufts that turned into pruned vessels after treatment, or a flow area associated with a smaller lesion, similar to those that Hwang described similar to MA [127] or to type 1 NVE described by Pan [121]. In 2018, Pan and colleagues studied NVD origins and found that it could originate either from the retinal artery or vein or from the choroid. They provided the most comprehensive classification of NVE types: type 1, the most frequent type, which arises from veins of the superficial plexus, and after reaching the posterior hyaloid, it branches forming a tree-like shape; type 2 is born from capillaries of deep vascular layers and presents as an octopus-like structure at the ILM; type 3 originates from veins located between the inner nuclear layer (INL) and GCL and creates sea-fan-like IRMAs [121]. These NVE types arise from capillary nonperfusion areas or close to them [121]. This venous origin contravenes previous studies where arteries were considered the origin of NVEs using FA [128].

In diabetic patients, OCTA is an effective tool to recognise poorly perfused or ischaemic areas at the margins of which NV is thought to arise both at the level of the optic disc, which has been most studied to date and has been related to a larger area of nonperfused retina, and other locations, despite being more frequent according to some studies [125]. Moreover, although the presence of NV is linked with diabetic retinopathy per se, several demographic factors, such as male sex and black ethnicity, are related to larger areas of NV [129].

OCTA imaging has shown clear advantages over traditional systems for NV. Nevertheless, recent studies have proven that WF-OCTA) detects more NV areas than conventional OCTA [129]. In this sense, it is important to define the best protocol to detect most vascular alterations in retinal areas. Most of these lesions are located at the posterior pole or mid-periphery of the retina, and 12 × 12° scans centred at the fovea and optic nerve and 15 × 9 scans are the most useful scans to localise NVE [129]. Although no differences were found between these scans, the 15 × 9 scan showed a greater number of artefacts [129]. Specifically, the detection rate of NV with 15 × 9 scans was 34.6%, compared with the detection rate of 17.6% using a 6 × 6 scan [129].

Thus, WF-OCTA has been proposed to be the only test necessary for the diagnosis and follow-up of NV because, taking FA as a reference, it has been demonstrated that fovea-centred WF-OCTA is able to reveal between 99.4% and 99.7% of NV [125]. Hirano and colleagues, using WF-OCTA with vitreoretinal interface segmentation, were able to detect NV in 84% of cases after manual segmentation. Automated segmentation with their devices had a 16% false-positive rate that diminished due to segmentation errors but was able to find nine NVs undetected with FA because of their small size [130]. Papayannis et al., using three new vitreo-retinal segmentation protocols with a Triton device, found a sensitivity and specificity in detecting NVD and NVE of 100% and 96.6%, respectively [124]. They used these new protocols to assess the activity of the NV.

On the other hand, Ishibazawa et al. observed and quantified the vascular changes (vascular density, ischaemia in the different plexuses) in NV and possible changes in the disc but did not confirm a correlated structural alteration [21,55,71]. Other changes in patients with NV were FAZ enlargement in both the SCP and DCP and/or nonperfusion areas [127].

#### 3.1.6. Diabetic Macular Oedema

DME is the main cause of vision loss in patients with DR. Macular cysts are visualised on OCTA as hyporeflective areas devoid of capillaries or flow signals with smooth borders [60,131]. They are located in the deep layers of the neurosensorial retina [131]. Some concerns exist about the reliability of OCTA to visualise DCP in DME. The absence of capillaries in the cysts of both plexuses may be secondary to the displacement of capillaries at the periphery of the cysts or to the preferential development of cysts in nonperfusion areas [19,132]. There are other possible factors related to the vascular changes in DME, such as the fluid attenuating the decorrelation signal from surrounding capillaries, the cyst exerting mechanical pressure on the vessel, or the capillaries being incompetent, leading to DME [23]. De Carlo et al., in 17 eyes, described the differences between cysts, with an oblong shape and smooth borders, devoid of flow and capillary nonperfusion with irregular borders and greyer hue [133].

Eyes with DR and DME are associated with reduced VD in OCTA compared with those with DR without DME [55,57]. They also present lower VD in the SCP and decreased perfusion of the DCP, revealing a more significant effect of oedema on macular perfusion at the level of the DCP and greater macular ischaemia at the deep retinal layers [55,132]. Kim and colleagues found different data depending on the DR grade. Diabetic patients with mild NPDR, with and without DME, showed that those with DME (8/32) had a lower VD, skeletal density, and fractal dimension in both superficial and deep retinal layers (60% of the inner retina vs. 40% of the outer retina) with a higher vessel density index in the deep retina layers [23]. Severe NPDR with DME (13/16) showed only a greater VD index in the deep retina layer, and patients with PDR with and without DME (24/36) showed no differences in the studied parameters [23]. Ting et al. also found a diminution in capillary density index in DM2 patients with DME in both the SCP (0.344 vs. 0.347) and DCP (0.349 vs. 0.357), but these differences did not reach statistical significance [108]. Mane and colleagues, studying 24 eyes with chronic diabetic cystoid macular oedema, described that the cysts were surrounded by capillary dropout areas in 71% and 96% of the cases in the SCP and DCP, respectively, with a diminished VD [134]. Sun et al. studied OCTA biomarkers for the progression of DR or development of DME. Patients with lower VD in the SCP were at higher risk of developing DME [55]. In DME, there is an imbalance between the liquid entering and exiting the retina. The leakage could proceed from the SCP, but the Müller cells and the DCP may be involved in removal. The DCP is the main venous outflow system, and its damage could generate DME [135].

Samara et al. [51] determined a significant enlargement of the FAZ area in diabetic eyes with DME at both the SCP and DCP, compared with the control group, and at the SRL when compared with diabetic eyes without DME. As previous studies, such as that of Balaratnasingam et al. [49], have observed, a significant correlation between FAZ area and VA in diabetic eyes with macular oedema existed with decreased VA in the larger FAZ area at both the superficial and deep retinal plexus [132]. Di et al. also described a larger FAZ in patients with DME than in DM patients without DME [33].

Additionally, VD at the SRL could be a predictive tool for VA in diabetic eyes with DME; a significant negative correlation is observed between VD at the SRL and LogMAR VA [132].

Spaces are surrounded by capillary nonperfusion, which shows no evidence of reperfusion after the resolution of DME [19], suggesting that DME might preferentially develop in areas of ischaemia. Mane and colleagues also described that after DME resolution, capillary density remained almost the same without reperfusion [134]. The same findings were reported by Ghasemi Falavarjani et al., in 13 DME patients after a single intravitreal injection, with no changes in capillary density or FAZ area [136]. Lee and colleagues described the response to anti-VEGF treatment. DM patients who did not respond to anti-VEGF therapy were those with damage to the integrity of the DCP but not the SCP, including lower flow density, larger FAZ, and a higher number of MAs [98]. They discussed the mechanism between the decrease in flow density in the DCP and the resistance to anti-VEGF treatment.

MAs in the DCP are thought to contribute to DME pathogenesis, with correlations between macular volume and MA density of the DCP [68]. They may also contribute to therapy, finding that the greater the MA proportion and the larger the FAZ area in the DCP, the worse the response to anti-VEGF therapy [137].

Some studies, such as Sun et al. [55], postulate that OCTA metrics provide independent risk information on microvasculature and could improve predictive discrimination for both DR progression and DME development compared with traditional, established risk. In their 2-year follow-up study, although they found changes in the DCP VD and FAZ, DME development was related to the VD of the SCP. However, OCTA metrics of the DCP were related to DR progression. They pointed out some limitations in their study but highlighted the role of OCTA metrics.

Further studies are needed to elucidate whether DME vascular changes are secondary to oedema and other OCTA risk biomarkers for DME.

#### 3.1.7. OCTA, DR and Deep Learning

There are significant differences between current multimodal devices and image processing methods, and reference ranges have not been established. Thus, some authors use artificial intelligence (AI), including deep learning (DL), to evaluate OCTA images; this is a machine learning technique which learns representations of data based on computational models with more efficient and precise results, and has already been applied to other ocular conditions [138]. In fact, the combination of AI models using OCT, OCTA and multimodal images appears to be more precise to detect changes in diabetic patients than the OCT AI model [139].

Guo et al., used DL to detect the NPA [140] using a multi-scale feature extraction capability to segment them from OCTA 6 × 6 m^2^ images, with great specificity and sensitivity and excellent performance (F1-score > 80%). This model was valid for different DR severities or image qualities (dice coefficient > 0.87) and was able to detect signal reduction artefacts [141].

Different DL-models assess OCTA image quality assessment [142], object segmentation [143], and quantification [144], with high accuracies. Ryu et al. developed a convolutional neural network (CNN) model classification algorithm with a sensitivity of 86–97%, a specificity of 94–99%, and an accuracy of 91–98% to diagnose DR through OCTA [145]. Le et al.’s DL classifier differentiated among healthy, no DR, and DR eyes with 83.76% sensitivity, 90.82% specificity, and an 87.27% accuracy [146] and Heisler et al. achieved an accuracy of between 90% and 92% [147]. Other DL techniques have shown an AUC of 0.91 to differentiate diabetic patients without DR from those with DR and an AUC of 0.8 to diagnose DR from non-diabetic patients [139]. This AUC was increased up to 92.33% in the DL model of Alam et al. to distinguish controls from DR [148].

Nazir et al.’s DL study was able to identify different severities of the DR based on local tetragonal OCTA patterns [149]. Hwang et al., suggested that the automated quantification of non-perfusion areas using projection-resolved OCTA is able to distinguish levels of DR [87]. Nagasawa and colleagues used a combination of UW fundus ophthalmoscopy and OCTA to stage DR [150]. They obtained a DL algorithm with sensitivities of 78.6% and 80.4% and specificities of 69.8% and 96.8% to distinguish NDR and DR and NPDR from PDR, respectively. These high percentages were, in part, related to Optos accuracy.

Xiong et al. compared commercial software measuring extrafoveal vessel density (EVD) with a DL-based macular extrafoveal avascular area (EAA) on 6×6 mm OCTA and demonstrated a better DR severity diagnostic accuracy; the results seem to be less conditioned by the signal strength and shadow artefacts [151]. Moreover, AI enabled the obtaining of OCTA images with less noise in order to analyse different vascular parameters more correctly, such as vessel density or fractal dimension [152]. Alam et al. also analysed several vascular parameters, and the algorithm that combined all of them reached 94.45% sensitivity, 92.29% specificity, and 92.96% accuracy in identifying mild NPDR with respect to the controls [38]. Among them, the vessel density obtained the best sensitivity to detect DR (with an accuracy of 93.89%) [38]. Otherwise, Detectron2, a new DL model, accurately measured the FAZ in diabetic eyes in a similar way to manual measurements [18].

Classifications of different stages of DR with OCTA DL models have obtained values of AUC of 0.865. Nevertheless, the combination of both OCT and OCTA images, and clinical and demographic data, reached the best AUC (0.96) [148]. The use of this new technique provides the possibility of early detection and may help in DR progression assessments with great accuracy and reliability, evaluating large amounts of data in a short time and reducing human labour, playing a key role in OCTA image analysis of this developing pathology. Multi-ethnic individuals with millions of samples are required to train DL-OCTA devices.

## 4. Summary and Conclusions

OCTA can provide a large number of findings about the retinal capillary layers and CC in DM patients even without signs of DR. OCTA offers advantages over FA, because it is a non-invasive and faster assessment that can be used as a routine exploration. A variety of metrics can be obtained, including FAZ, acircularity index, axis ratio, VD, NV, and other vascular parameters, such as fractal dimension, vessel tortuosity, or skeleton density, that can be considered markers of the disease and progression. Due to variability between subjects, OCTA results differ from one study to another.

We performed a systematic review of the studies published in this field. We checked for other papers missing after the review process, but some information could have been missed.

Development and improvements in OCTA devices, such as the protocols, the studied field, acquisition speed, and automatically performed measurements including FAZ measurements and irregularity, density, and flow index, and other quantitative features including blood vessel calibre, tortuosity, vessel branching coefficient, and angle, etc., are important clinical benefits in the diagnosis and control of both preclinical and clinical DR. It will offer great advantages in detecting vascular changes, NPA, or NV, not only at the macula, but also at the periphery with WF-OCTA, which could change the diagnosis and disease management.

Several studies have detected potential OCTA biomarkers for DR development or for treatment response. OCTA with multimodal images and systemic biomarkers may guide follow-up and treatment options as well as vascular changes after treatment response. More studies are needed to address the importance of all these factors.

## Figures and Tables

**Figure 1 biomedicines-10-00088-f001:**
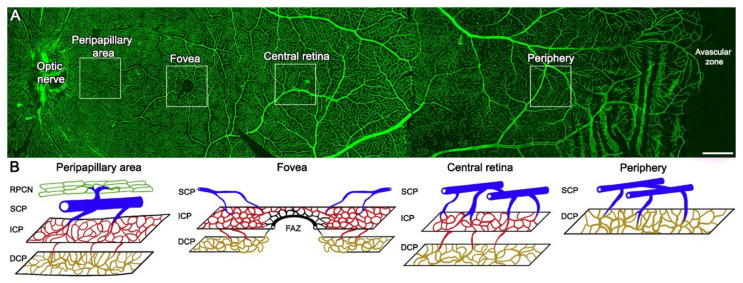
Morphological changes in the vascular plexuses throughout the human retina. (**A**) Whole-mount human retina immunostained with an antibody against collagen type IV showing the vascular network from the optic nerve to *Ora*
*serrata*. (**B**) Drawings of the different plexuses corresponding to the insets in (**A**). Four plexuses can be observed in the peripapillary area (RPCN, SCP, ICP, and DCP) close to the optic nerve. The central retina is composed of three plexuses (SCP, ICP, and DCP), except in the fovea where the foveal avascular zone (FAZ) exists. Only two plexuses (SCP and DCP) are present in the far-periphery area. RPCN, radial peripapillary capillary network; SCP, superficial capillary plexus; ICP, intermediate capillary plexus; DCP, deep capillary plexus. Scale bar: 1 mm.

**Figure 2 biomedicines-10-00088-f002:**
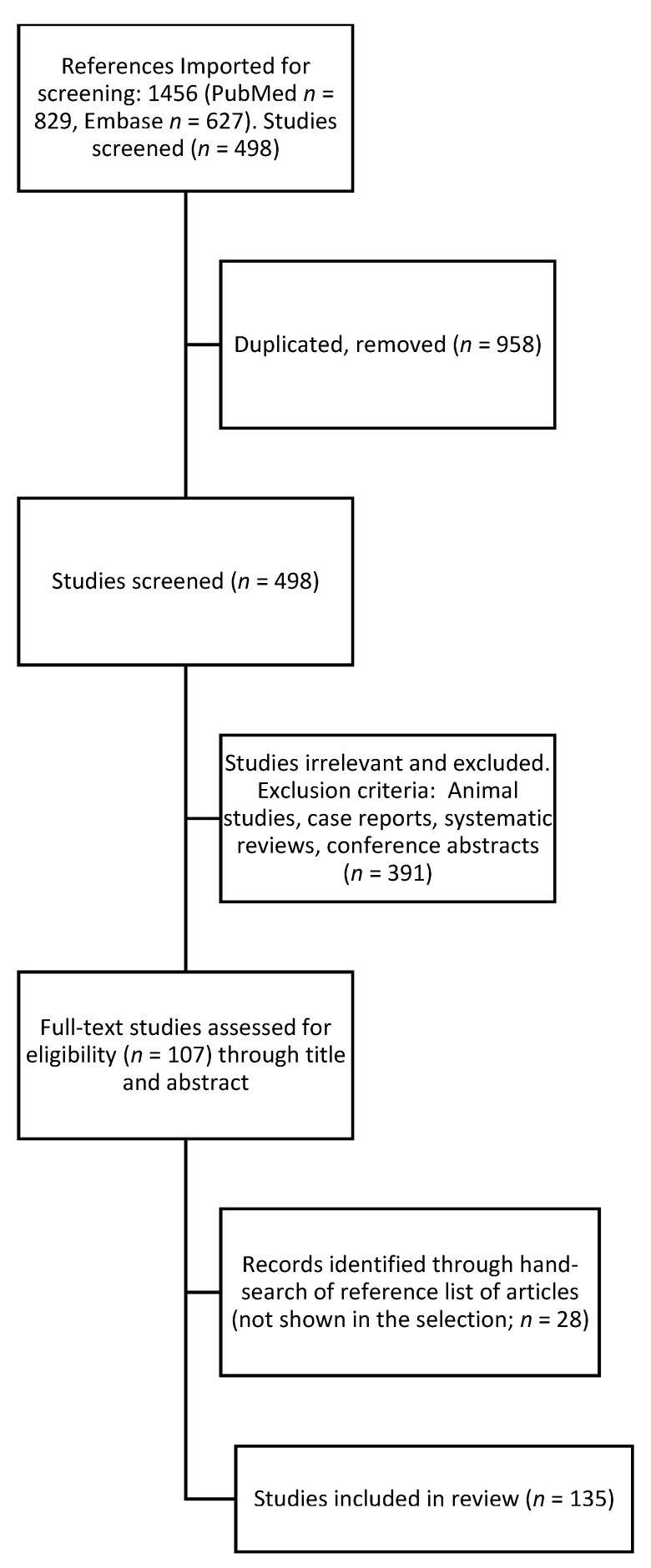
Flow chart explaining the literature selection. A systematic review was performed following PRISMA guidelines. A total of 1456 records were selected (829 in PubMed and 627 in Embase), and after the removal of duplicate studies or articles that did not meet the selection criteria, 107 articles were selected for a full literature review. Ultimately, a total of 135 studies were included after adding important works that were not found in the databases.

**Figure 3 biomedicines-10-00088-f003:**
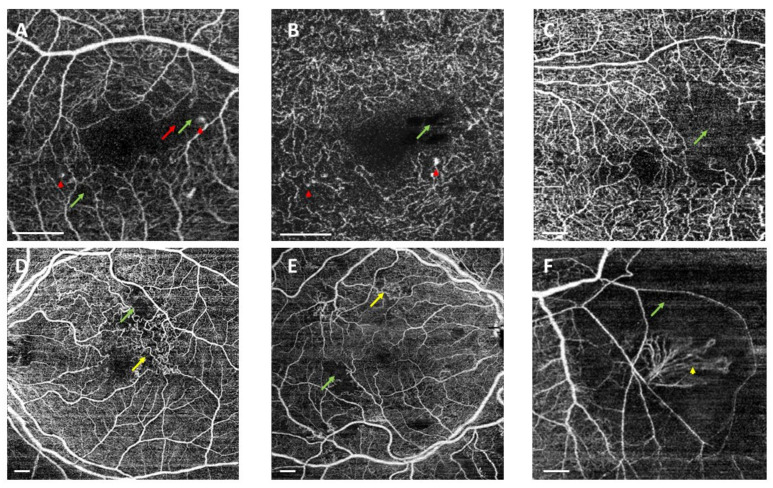
Swept source optical coherence tomography angiography (SS-OCTA) showing representative examples of OCTA findings in diabetic patients. OCTA was acquired using DRI-Triton SS-OCT (Topcon, Tokyo, Japan). (**A**,**B**) Superficial (SCP) and deep capillary plexuses (DCP) in 3 × 3 mm scans. Figure 3C shows SCP in a 6 × 6 scan. (**D**,**E**) The SCP in a 9 × 9 scan and 3F SCP in a 6 × 6 scan protocol. (**B**) An irregular foveal avascular zone (red arrows) and microaneurysms (red arrowheads) in the superficial and deep capillary plexuses. (**C**) Nonperfusion areas in the temporal area (green arrows). (**D**,**E**) Areas of impaired perfusion associated with intraretinal microvascular abnormalities (yellow arrows). (**F**) Retinal neovascularization elsewhere (yellow arrowhead). Scale bar represents 1 mm.

**Figure 4 biomedicines-10-00088-f004:**
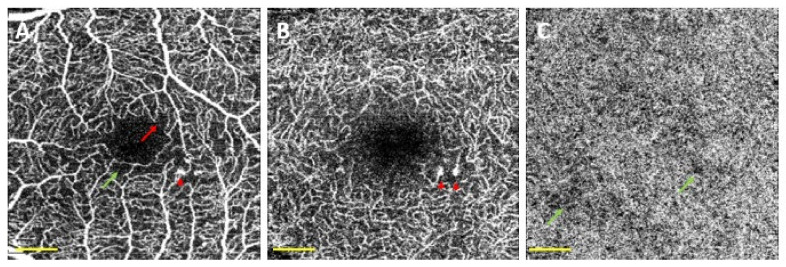
Optical coherence tomography angiography (OCTA) of a diabetic patient without detectable diabetic lesions. OCTA was acquired using DRI-Triton SS-OCT (Topcon, Tokyo, Japan) with a 3 × 3 mm scan protocol. (**A**) Superficial capillary plexus, (**B**) deep capillary plexus, and (**C**) choriocapillaris (CC). Red arrow shows a disruption in the foveal avascular zone (FAZ), red arrowheads correspond to microaneurysms, and green arrows correspond to non-perfusion areas in the CC. Scale bar (in yellow) represents 500 microns.

**Figure 5 biomedicines-10-00088-f005:**
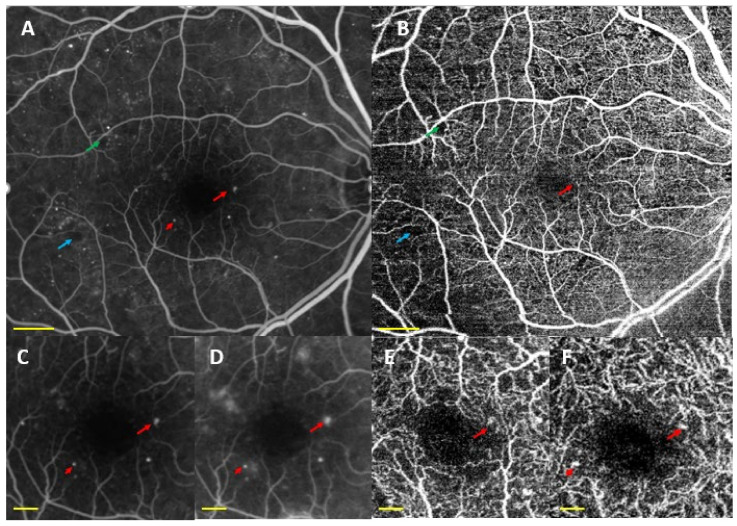
Fluorescein angiography (FA) (**A**,**C**,**D**) vs. swept source optical coherence tomography angiography (SS-OCTA) (**B**,**E**,**F**) showing microaneurysms (MAs) (red arrow and arrowhead) in diabetic patients. (**A**,**C**,**D**) MAs in the superficial and deep capillary plexuses (SCP and DCP) and nonperfusion areas (green and blue arrows) detected by (**A**,**B**,**E**,**F**) show the same MAs in SCP (**E**), DCP (**F**), and nonperfusion areas (**B**), visualised by OCTA. FA was acquired using a Spectralis-HRA (Heidelberg Engineering, Heidelberg, Germany). (**A**,**C**) Arterial time in the FA and (**D**) tissue times. OCTA was acquired with DRI-Triton SS-OCT (Topcon, Tokyo, Japan). (**B**) 9 × 9 mm OCTA and (**E**,**F**) 3 × 3 mm OCTA of both SCP and DCP, respectively. Scale bar (in yellow) represents 1 mm in Figure 5A,B and 250 microns in Figure 5C,F.

## Data Availability

Not applicable.

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
