# Peer review of "Optical Coherence Tomography Angiography in Diabetic Patients: A Systematic Review"

_biomedicines, 2021, doi:10.3390/biomedicines10010088_

Round 1

Reviewer 1 Report

Thank you for allowing me to review this work

Here are my comments

Abstract

  1. The reason for performing this study in the intro is missing. What made you start this work? What is the clinical question you want to answer?
  2. N=117- impressive
  3. This paper supports the use of OCTA in the conclusion part, but no results are given for this in the results part

Intro

  1. 35% of DR in DM is a significant figure
  2. The intro is very long-some parts could be omitted or transferred to the discussion.
  3. For example, 1.1 does not see how this is related here could be added as a supplemental
  4. Please elaborate more on the aim:” The purpose of this review was to provide an actual summary of the different findings assessed by OCTA and the diagnostic value of OCTA in DR patients.” do you have any problems diagnosis now DR? is there any case you miss patients?

Method

  1. Prisma- well done- how many authors performed this search?

Results:

  1. This is a descriptive report, not a quantifiable analysis- I would mention this in the title/abstract. This is a profound, in-depth review of the literature- not sure how much the average clinician doctor could understand from this- can you chunk it up? Summerize it?

Conclusion:

  1. OCTA can provide a large number of findings in the retinal capillary layers and CC in DM patients even without signs of DR.?” can you provide images of these findings?
  2. “It offers an advantage over FA being a non-invasive and faster assessment that can be used as a routine exploration” we knew this even before your review- what is new here? What is this study add? Would you please explain why a reader should spend the time going over this study? How will it benefit the new DM patient coming to the clinic for evaluation?
  3. Several studies detect potential OCTA biomarkers for RD development or treatment response. ---did you mean DR?

Overall-

This English language of this study is extra-ordinary – well done

This is a comprehensive review, and significant effort has been made here- I believe this work merits a publication. I would encourage you to try to link this work to a clinical setting, as I previously mentioned

Author Response

Dear Reviewer,

We appreciate your comments about the interest of this manuscript.

We revised the manuscript according to your suggestion and all the change included in the manuscript is detailed below.

We proceed to answer the editorial and the reviewers’ comments point-by-point.

Abstract

  1. The reason for performing this study in the intro is missing. What made you start this work? What is the clinical question you want to answer?

As suggested, we included the following sentence in the abstract background: ‘’The aim of this article is to analyze the current applications of OCTA and provide an updated overview of them in the evaluation of DR’’ (Pg 1, 1st paragraph, line 29)

  1. N=117- impressive

We sincerely appreciate your comments and interest in our manuscript. 

  1. This paper supports the use of OCTA in the conclusion part, but no results are given for this in the results part

Results section has been explained to clarify findings as suggested (Pg 1, 1st paragraph, line 38).

Intro

  1. 35% of DR in DM is a significant figure

We sincerely appreciate your comments and interest in our manuscript. 

  1. The intro is very long-some parts could be omitted or transferred to the discussion. For example, 1.1 does not see how this is related here could be added as a supplemental

As suggested, introduction was summarized and part 1.1 was replaced as ‘’Supplementary material 1’’ (Pg 3, 7th paragraph, line 92).

  1. Please elaborate more on the aim:” The purpose of this review was to provide an actual summary of the different findings assessed by OCTA and.” do you have any problems diagnosis now DR? is there any case you miss patients?

As suggested, we have improved the aim of our study as follows: ‘’ …the diagnostic value of OCTA in DR patients, which is a great future challenge due to DM prevalence and the heavy burden caused by DR’’ (Pg 4, 6th paragraph, line 142).

Method

  1. Prisma- well done- how many authors performed this search?

PRISMA search was performed by three authors: two independent reviewers started initial review of abstracts, removal of duplicate studies or those articles that did not meet the selection criteria, and another author help in the selection of other papers that didn´t show in the selection which appear to be important to our review. These points have been addressed at the methods (Pg 5, 3th paragraph, line 162).

Results:

  1. This is a descriptive report, not a quantifiable analysis- I would mention this in the title/abstract. This is a profound, in-depth review of the literature- not sure how much the average clinician doctor could understand from this- can you chunk it up? Summerize it?

As suggested we have summarize the review and add a supplement to make understand easier for the reader (Pg 3, 7th paragraph, line 92).

Conclusion:

  1. OCTA can provide a large number of findings in the retinal capillary layers and CC in DM patients even without signs of DR.?” can you provide images of these findings?

Images of retinal capillary plexuses and choriocapillaris findings in DM patients without signs of DR have been included (Figure 3 – Pg 8, 2nd paragraph, line 216).

  1. “It offers an advantage over FA being a non-invasive and faster assessment that can be used as a routine exploration” we knew this even before your review- what is new here? What is this study add? Would you please explain why a reader should spend the time going over this study? How will it benefit the new DM patient coming to the clinic for evaluation?

We provide an actual summary of the different findings assessed by OCTA and the diagnostic value of OCTA in DR patients and discuss the novelty use of automatic (or semi-automatic) image analysis methods in DR diagnostics (Pg 20, 5th paragraph, line 733)

  1. Several studies detect potential OCTA biomarkers for RD development or treatment response. ---did you mean DR?

The highlighted spelling mistake was modified to ‘’DR’’ instead of ‘’RD’’.

Best regards,

Ana Boned Murillo

Isabel Pinilla, MD PhD

Reviewer 2 Report

Dear Authors,

Manuscript ID: biomedicines-1507235 entitled, "optical coherence tomography angiography in diabetic patients," is compelling. The authors intended to review the OCTA to evaluate changes in DM patients with or without DR in writing this manuscript. The thorough discussion of each change that OCTA can detect is well-described. However, supplementing more clinical photographs of OCTA findings in each change could have made this review engaging and thorough. Even though the references in the manuscript have missed some of the recent studies on OCTA, the data they provide is still informative and adequate. The draft is comprehensive, with a simple narrative structure. Some of my suggestions include:

  • Even the manuscript is well-written, the manuscript requires grammatical correction and professional English editing. There are a few errors to mention here:
  • In the abstract, Please correct making "developed" countries. Is "selection criteria removal" structured correctly in the sentence? Please correct to make "provided" by OCTA. Include a full form of FAZ in the abstract.
  • In Section 1: please correct to make "developed" countries. Is the "thickness increase" of the basal membrane is structurally making sense?
  1. The non-invasive nature of the method alone cannot be taken as a comparative advantage over an invasive in clinical diagnosis. Fluorescein angiography (FA) and indocyanine green angiography (ICGA) are standard imaging modalities to visualize blood vessels and the dynamic changes within the retinal vasculature. Would you please tabulate and describe the significance of OCTA over FA and ICGA briefly?
  2. Wherever possible, please include a clinical photograph for seven changes assessed by OCTA compared to FA for making the review more engaging and understandable.
  3. "RD" is mentioned without full form in some manuscript paragraphs. Is it mistaken by diabetic retinopathy (DR)? or do provide a full form for RD. As in Section 4, several studies detect potential OCTA biomarkers for the "RD development" or for treatment response.

Author Response

Dear Reviewer,

We appreciate your comments about the interest of this manuscript.

We revised the manuscript according to your suggestion and all the change included in the manuscript is detailed below.

We proceed to answer the editorial and the reviewers’ comments point-by-point.

  1. Even the manuscript is well-written, the manuscript requires grammatical correction and professional English editing.

Professional English editing has been performed as suggested.

  1. In the abstract, Please correct making "developed" countries.

The highlighted spelling mistake was modified to ‘’’developed’’ instead of ‘’develop’’ (Pg 2, 1st paragraph, line 24).

  1.  Is "selection criteria removal" structured correctly in the sentence?

The highlighted spelling mistake was modified to ‘’’…that duplicate articles and those which did not meet the selection criteria were removed’’ instead of ‘’…selection criteria removal’’ (Pg 2, 2nd paragraph, line 36).

  1. Please correct to make "provided" by OCTA.

This sentence was removed from original text.

  1.  Include a full form of FAZ in the abstract.

The full form ‘’Foveal avascular zone’’ was included in the abstract as suggested (Pg 2, 3th paragraph, line 41).

  1. In Section 1: please correct to make "developed" countries.

The highlighted spelling mistake was modified to ‘’developing’’ (Pg 3, 2nd paragraph, line 63).

  1. Is the "thickness increase" of the basal membrane is structurally making sense?  

Basal membrane thickness and loss of pericites are one of the main reason of the ischemia and leakage in DR. OCTA could detect ischemia secondary to this change but its resolution and the way of detect vessel based on cell movement is not going to detect neither of both alterations.

  1. The non-invasive nature of the method alone cannot be taken as a comparative advantage over an invasive in clinical diagnosis. Fluorescein angiography (FA) and indocyanine green angiography (ICGA) are standard imaging modalities to visualize blood vessels and the dynamic changes within the retinal vasculature. Would you please tabulate and describe the significance of OCTA over FA and ICGA briefly?

OCTA data acquisition is faster than FA and is three dimensional and depth resolved, allowing individual capillary plexuses automatically assessment based on current software algorithm. OCTA allows the visualization of all plexuses, including the intermediate capillary, detecting pathological features that are not available in traditional dye-based angiography. Besides, as a non-invasive and fast test it is adequate for patients who require frequent follow-up exams. Nevertheless, FA is still the gold standard for retinal vessels evaluation, providing some additional findings such as leakage (Pg 6, 3th paragraph, line 185).

  1. Wherever possible, please include a clinical photograph for seven changes assessed by OCTA compared to FA for making the review more engaging and understandable.

As suggested, we have included some significant FA photographs compared to OCTA images previously exposed (Figure 4 - Pg 10, 2nd paragraph, line 299).

  1. "RD" is mentioned without full form in some manuscript paragraphs. Is it mistaken by diabetic retinopathy (DR)? or do provide a full form for RD. As in Section 4, several studies detect potential OCTA biomarkers for the "RD development" or for treatment response.

The highlighted spelling mistake was modified to ‘’DR’’ in the text.

Best regards,

Ana Boned Murillo

Isabel Pinilla, MD PhD

Reviewer 3 Report

The paper presents diagnostic applications of OCT angiography in diabetic retinopathy. This work is an interesting review of many studies that use OCTA to analyze the pathology of retinal vascularization or changes in vascular perfusion. Unfortunately, in the current version this paper is not novel enough to be published. The introduction contains two redundant sections ("Morphology of blood vessels of the retina and choroid", "Optical coherence tomography angiography") which cover well-known topics. Also, the main conclusion is obvious - it is well known that OCTA is a reliable tool for diagnosing and predicting DR.

I recommend extending this work with a chapter that discusses the use of automatic (or semi-automatic) image analysis methods in DR diagnostics. There are many studies in which artificial intelligence techniques prove to be an effective tool to aid in the diagnosis of vascular changes in the retina, because such techniques can analyze salient features extracted from OCTA images. Currently, there is no state-of-the-art analysis in this area. This chapter will therefore be an important novelty of this work and should arouse considerable interest of the readers.

Author Response

Dear Reviewer,

We appreciate your comments about the interest of this manuscript.

We revised the manuscript according to your suggestion and all the change included in the manuscript is detailed below.

We proceed to answer the editorial and the reviewers’ comments point-by-point.

  1. Introduction contains two redundant sections ("Morphology of blood vessels of the retina and choroid", "Optical coherence tomography angiography") which cover well-known topics

As suggested, Introduction has been summarized. Part of the text has been added as supplementary material #1 (Pg 23, 4th paragraph, line 825).

  1. I recommend extending this work with a chapter that discusses the use of automatic (or semi-automatic) image analysis methods in DR diagnostics. This chapter will therefore be an important novelty of this work and should arouse considerable interest of the readers.

As suggested, a new chapter entitled “OCTA, DR and deep learning” has been included in the review (Pg 20, 5th paragraph, line 733).

Reviewer 4 Report

The authors present an interesting systematic review analysing the rapid rise of optical coherence tomography angiography (OCTA) in the context of visual diagnostics. Ophthalmic diseases are rapidly on the rise in today’s age owing to the increase in incidence of diseases such as diabetes that increase the risk of vision-associated conditions and disease states. In order to ease the burden on healthcare systems, rapid, accurate, and less invasive means of profiling a patients ocular health are required, and the authors review the strengths of OCTA in the context of several ocular disease states. Overall, this is a detailed, well researched review, however, in reviewing the text I had a number of concerns. The following should be addressed by the authors when preparing a suitable revision.

  1. The writing for the most part if good, but there is room for improvement. There are several typos, grammatical and formatting errors within that require attention. While the detail is good, the writing itself can make the flow quite disjointed, and while the point being made is understandable, it is not currently to publication standard. The authors should revise the entire manuscript and address these issues in advance of any resubmission.
  2. The referencing could be improved in certain instances. For example, in the introduction there are substantial amounts of information given with little to no supporting references to back them up. The authors should review this in advance of any resubmission.
  3. The authors include ophthalmic images as part of the study. What are the sources of these images? If they are provided by the labs of the authors it would be preferable if this was mentioned, with details on the origin of the image (Figure 1 and 3). Also a reference to how these images were acquired (reference the method) would also be useful. If the authors acquired this from another research group/article, then this should be referenced appropriately – at the moment it is not clear.  
  4. In Figure 2, the software package used to generate the image appears to have left overlays in the image – it appears to be a spellcheck red underline. This needs to be removed.
  5. It might be useful in Figure 3 to include arrows pointing at regions within the images to which the authors are referring to.
  6. It would be useful if some of the weaknesses of OCTA were included in the piece. This piece is very good at highlighting the strengths of the approach, but in a time when techniques are quickly evolving and becoming outdated or replaced, a ‘future perspectives’ or similar might give a more balanced take on the technique itself.

Author Response

Dear Reviewer,

We appreciate your comments about the interest of this manuscript.

We revised the manuscript according to your suggestion and all the change included in the manuscript is detailed below.

We proceed to answer the editorial and the reviewers’ comments point-by-point.

  1. The writing for the most part if good, but there is room for improvement. There are several typos, grammatical and formatting errors within that require attention. While the detail is good, the writing itself can make the flow quite disjointed, and while the point being made is understandable, it is not currently to publication standard. The authors should revise the entire manuscript and address these issues in advance of any resubmission.

Professional English editing has been performed as suggested.

  1. The referencing could be improved in certain instances. For example, in the introduction there are substantial amounts of information given with little to no supporting references to back them up. The authors should review this in advance of any resubmission.

New references have been added to the text to support given information (references 2, 5, 6, 9 and 10).

  1. The authors include ophthalmic images as part of the study. What are the sources of these images? If they are provided by the labs of the authors it would be preferable if this was mentioned, with details on the origin of the image (Figure 1 and 3). Also a reference to how these images were acquired (reference the method) would also be useful. If the authors acquired this from another research group/article, then this should be referenced appropriately – at the moment it is not clear.

The images are from our clinic; that is the reason that we haven’t said the procedure. We will clarify the issue in the figure legend.

  1. In Figure 2, the software package used to generate the image appears to have left overlays in the image – it appears to be a spellcheck red underline. This needs to be removed.

Red line has been removed (Figure 2).

  1. It might be useful in Figure 3 to include arrows pointing at regions within the images to which the authors are referring to.

We have included arrows and arrowshead for pointing up the lesions (Figure 3).

  1. It would be useful if some of the weaknesses of OCTA were included in the piece. This piece is very good at highlighting the strengths of the approach, but in a time when techniques are quickly evolving and becoming outdated or replaced, a ‘future perspectives’ or similar might give a more balanced take on the technique itself.

OCTA limitations have been described in introduction section (Pg 4, 4th paragraph, line 129) and future perspectives were analyzed in a new chapter entitled “OCTA, DR and deep learning” (Pg 20, 5th paragraph, line 733).

.

Best regards,

Ana Boned Murillo

Isabel Pinilla, MD PhD

Round 2

Reviewer 2 Report

Dear Authors,

Thank you for taking the time to improve the draft based on suggestions. The addition of clinical photographs (OCTA and FA) significantly enhances the review's interpretability. Some minor comments:

  1. In figure 1 flowchart, correct to make "PubMed" instead of "Pubmed".
  2. I highly recommend moving the supplemental figure (morphology) in the introduction section and describing it briefly. OR remove instead of keeping it in the supplementary material.

Than you

Author Response

Dear Reviewer,

We revised the manuscript according to your suggestion and all the change included in the manuscript is detailed below.

Thank you for taking the time to improve the draft based on suggestions. The addition of clinical photographs (OCTA and FA) significantly enhances the review's interpretability.

Thank you for helping us to improve the paper

Some minor comments:

In figure 1 flowchart, correct to make "PubMed" instead of "Pubmed".

We have corrected the mistake

I highly recommend moving the supplemental figure (morphology) in the introduction section and describing it briefly. OR remove instead of keeping it in the supplementary material.

We moved back the figure to the manuscript and leave the explanation the rest of the explanation in the supplementary material

Thank you again

Isabel Pinilla, MD PhD

Reviewer 3 Report

Thank you for addressing my comments.

Just two minor issues needs clarification:

line 736 "authors use artificial intelligence (AI) or deep learning (DL) to evaluate OCTA images" - in fact, deep learning is subset of AI methods thus please modify: "authors use artificial intelligence (AI), including deep learning (DL) to evaluate OCTA images"

line 739: this phrase is not clear - which models more precise detect changes in the diabetic patients? What is crucial, combination of OCT and OCTA or application of specific AI models? Please clarify.

Then the paper will be suitable for publication.

Author Response

Dear Reviewer,

We appreciate your comments about the interest of this manuscript.

Thank you for addressing my comments.

Thank you for your help

Just two minor issues needs clarification:

line 736 "authors use artificial intelligence (AI) or deep learning (DL) to evaluate OCTA images" - in fact, deep learning is subset of AI methods thus please modify: "authors use artificial intelligence (AI), including deep learning (DL) to evaluate OCTA images"

We corrected the phase following your suggestion

line 739: this phrase is 

Reviewer 4 Report

The authors have responded positively to my comments and for that the manuscript is much improved. 

Author Response

Thank you again for your suggestions

Isabel Pinilla, MD PhD